# Facing Multiple Environmental Challenges through Maximizing the Co-Benefits of Nature-Based Solutions at a National Scale in Italy

**Elena Di Pirro** [1], **Lorenzo Sallustio** [1], **Joana A. C. Castellar** [2,3], **Gregorio Sgrigna** [4], **Marco Marchetti** [1,*] **and Bruno Lasserre** [1]

1   Dipartimento di Bioscienze e Territorio, Università degli Studi del Molise, 86090 Pesche (IS), Italy; elena.dipirro@unimol.it (E.D.P.); lorenzo.sallustio@unimol.it (L.S.); lasserre@unimol.it (B.L.)
2   University of Girona, Plaça de Sant Domènec 3, 17004 Girona, Spain; jcastellar@icra.cat
3   Catalan Institute for Water Research (ICRA), Carrer Emili Grahit 101, 17003 Girona, Spain
4   Consiglio Nazionale delle Ricerche (CNR), Istituto di Ricerca sugli Ecosistemi Terrestri (IRET), URT UniSalento, Centro Ecotekne, Monteroni, Pal. B-S.P. 6 Lecce, Italy; gregorio.sgrigna@iret.cnr.it
*   Correspondence: marchettimarco@unimol.it

**Abstract:** The European Union is significantly investing in the Green Deal that introduces measures to guide Member States to face sustainability and health challenges, especially employing Nature-Based Solutions (NBS) in urban contexts. National governments need to develop appropriate strategies to coordinate local projects, face multiple challenges, and maximize NBS effectiveness. This paper aims to introduce a replicable methodology to integrate NBS into a multi-scale planning process to maximize their cost–benefits. Using Italy as a case study, we mapped three environmental challenges nationwide related to climate change and air pollution, identifying spatial groups of their co-occurrences. These groups serve as functional areas where 24 NBS were ranked for their ecosystem services supply and land cover. The results show eight different spatial groups, with 6% of the national territory showing no challenge, with 42% showing multiple challenges combined simultaneously. Seven NBS were high-performing in all groups: five implementable in permeable land covers (urban forests, infiltration basins, green corridors, large parks, heritage gardens), and two in impervious ones (intensive, semi-intensive green roofs). This work provides a strategic vision at the national scale to quantify and orient budget allocation, while on a municipal scale, the NBS ranking acts as a guideline for specific planning activities based on local issues.

**Keywords:** human health; human well-being; urban sustainability; green deal; urban forests; green roofs; multifunctionality

## 1. Introduction

In the European Union (EU), air pollution and the extreme events related to climate change (e.g., heatwaves and floods) are exerting pressure both on human health and natural capital integrity [1], leading to millions of premature deaths and economic losses each year [2]. This is especially relevant in urban areas, where 73% of the European population lives, compared to 50% globally [3,4]. For this reason, the EU is significantly investing in the European Green Deal, which introduces legislative and non-legislative measures to legally bind and guide the Member States to face sustainability and health challenges. The EU fixed targets across different strategies (e.g., Forestry and Biodiversity Strategy to 2030), laws (e.g., European climate law), and action plans (e.g., zero pollution action plan) [5] that the Member States need to meet at the national level for improving the quality of ecosystems and human life [6]. For example, a recent study by Khomenko et al. [7] estimated that about 52,000 lives would be saved annually if 1000 European cities met World Health Organization (WHO) air-quality standards. Particular attention is paid to

policies and planning at the local scale to reconfigure urban areas so that they consume fewer resources, generate less pollution (including greenhouse gases), and become more resilient and sustainable [4] while facing budgetary pressure [8].

As a consequence, there is a growing interest in valuing Ecosystem Services (ES) and including them in decision-making processes [1,9] as a lens to achieve environmental and societal goals [10]. Hence, the concept of Nature-Based Solutions (NBS) rises as an environmentally friendly alternative to favor the provision/maintenance of ES. NBS are defined as "solutions that are inspired and supported by nature, which are cost-effective, simultaneously provide environmental, social and economic benefits and help build resilience" [2]. NBS is an umbrella concept related to and integrated into other concepts, such as green and blue infrastructure, urban forestry, ecological engineering, disaster risk reduction, and ecosystem-based adaptation [11–14]. These concepts were introduced to address the challenges from distinct perspectives while the NBS strength is the integrated perspective for providing co-benefits and generating win–win solutions (i.e., multifunctionality) [13,15,16]. Moreover, implementing NBS can foster both human well-being and biodiversity cost-effectively while offering new job and innovation opportunities [17].

Therefore, both governmental and non-governmental organizations are offering huge funding globally [18,19] to enable the implementation of NBS [20,21]. The focus is predominantly on afforestation and reforestation programs [22], e.g., "3 billion trees" in the EU [23], the "trillion tree campaign" [24], and the "great green wall" [25]. Notwithstanding, McDonald et al. [26] highlight that funds for tree-planting and maintenance initiatives are often constrained or limited by planning silos. Indeed, governments generally receive several indications on NBS from the EU that are not easily translated into effective and practical urban greening programs. For this purpose, the Horizon 2020 program classified NBS as a priority area of investment to enhance the resilience in urban areas in the face of global changes [1] and establish Europe as a world leader of NBS [10,27]. Demonstration projects of NBS—and related concepts—are in place in several cities of the Member States to tackle different urban issues such as the mitigation of air pollution, temperature extremes, noise, drought, and flooding [27–30]. EU-funded projects on NBS work in task forces to improve knowledge, reduce duplication, and facilitate progress towards shared goals [31]. These projects are proving to be a catalyst for research-practice partnerships [18], increasing knowledge and awareness regarding NBS indicators, impacts, performance, and cost-effectiveness assessment, building repositories with different case studies (e.g., OPPLA, the online EU repository of NBS). All projects aim at strengthening NBS regional development and translating results from experts to stakeholders [6]. More details regarding the status of H2020 projects are available from Wild et al. [31].

However, projects are still often implemented as standalone experiments in urban areas, scattered and uncoordinated throughout various policy levels and sectors [32,33]. As hubs of population and socio-economic activity, urban areas represent concentrated opportunities for addressing issues of sustainability at the local scale [4,18]. Nevertheless, ameliorating the environmental conditions in a few cities can only partially contribute to delivering the national-level commitments that countries have with the EU and with United Nations [33]. Therefore, lessons learned from single case studies need to be coordinated across multiple political and geographical levels to enable the long-term respect of national targets and international commitments [34,35]. Despite the fact that NBS are implicitly or explicitly cited in different European legislative frameworks [6,36,37], the H2020 NATURVATION project underlined as a legal initiative or policy coordination at the EU level requiring Member States to systematically program and invest in NBS is still absent [32]. In addition, the review conducted by Mendonça et al. [20] reveals that the policy instruments to mainstream the NBS concept into policy are usually investigated just at the city level, thus neglecting the country or higher levels of implementation.

However, considering the huge funding opportunity for the member states envisaged by the EU Green Deal—and other key policy initiatives [37]—national governments need to develop appropriate strategies to coordinate local projects and face multiple and complex

challenges throughout the territory [33] to maximize the NBS effectiveness [22,32] and involve all relevant stakeholders [9,38]. Although this strategic level is still missing in most of the member states [32], it is crucial, especially for countries located in vulnerable areas currently facing climate and pollution issues (e.g., Mediterranean region [39]). In these countries, a wide and national perspective could help to coordinate the implementation of NBS at lower levels for reaching multiple national targets related to different environmental policies, with the final scope to improve the state of ecosystems and human health as a whole. Accordingly, we selected Italy as a case study, since it is a representative member state both for the challenges related to pollution and climate change and for the national policies in place to improve urban sustainability.

This paper thus aims to introduce a replicable methodology to integrate and strengthen NBS into a multi-level planning process to maximize their cost–benefit from the large-scale policy and planning initiatives (e.g., national) to the local scale (e.g., municipal). Generally, the former is focused on ameliorating environmental sustainability through reaching fixed targets, while the second is oriented to directly reconfigure urban areas, improving the wellbeing of inhabitants. Although many authors have already dealt with methods and approaches for planning and designing multifunctional NBS [40,41], they are usually limited to the municipal scale. Therefore, to the best of our knowledge, this is the first framework conducted on a national scale. We started from the Environmental Quality Standards set by Di Pirro et al. [42] to map the spatial co-occurrence of the same environmental challenges (individual or multiple) nationwide (i.e., spatial groups). These spatial groups of challenges serve as functional areas where NBS providing multiple co-benefits can be identified to effectively mitigate peculiar multiple environmental stressors altering human health and well-being (i.e., air pollution, heatwaves, flood hazards). Consequently, different rankings of 24 NBS were built for each spatial group based on i) their capacity to supply ES, and hence their performance to address the challenges, and ii) the current land cover in which they can be implemented. Using the Environmental Quality Standards helped to establish a replicable, clear, and spatially explicit understanding of the challenges that the country needs to tackle. The here-proposed framework is able to support the strategical coordination of national funds allocation and to enhance the effectiveness of interventions at the local scale consistent with the national objectives.

## 2. Case Study

In the Mediterranean basin, climate change has exacerbated existing environmental challenges caused by the combination of increasing pollution, land use changes, and declining biodiversity [39]. Indeed, Italy is consistently experiencing the adverse effects of climate change, such as heatwaves, floods, and drought events, combined with the strong exposition of the three most harmful air pollutants in the EU [39,43–46]. In addition to these challenges, within the Italian territory, the sealed surface reaches one of the highest relative national coverages (7.1% [47]) among EU countries [48]. The scattered and fragmented urban mosaic [49] has smoothed the boundaries between urban and rural areas [50], exacerbating issues related to ecological connectivity, biodiversity, and ecosystem services loss [51–54]. Therefore, the Italian national government envisaged different urban sustainability strategies and policies and set ambitious tree-planting objectives, based on the premise that planning urban forests is a feasible response to current challenges and that they can enhance the resilience of cities and safeguard the population's health [55]. For example, the "Decree on Climate" [56] is a national policy adopted to tackle the climate emergency and achieve the objectives related to the EU Air Quality Directive [57]. Within the decree, the "Urban Forestry Program" (Azioni per la riforestazione-Art.4 [56]) allocates funds to implement urban and peri-urban forests and to reduce impervious surfaces (i.e., de-sealing actions), as key interventions to address urban challenges [58]. The National Strategy on Urban Greenspaces [59] guides the municipalities to the effective implementation of local-scale initiatives for strengthening ecological networks. The Urban Forestry Program allocates funds to just 14 metropolitan cities, while the National Strategy

on Urban Greenspaces includes all the Italian municipalities in its analysis. However, Di Pirro et al. [42] reveal an incomplete spatial agreement between the current fund's allocation envisaged by the Urban Forestry Program and the real spatial distribution of the challenges to address. Sallustio et al. [60] highlight that the inclusion of all municipalities can ensure an equitable distribution of economic resources and provide guidelines that can be easily replicated and implemented at the municipal scale.

While the current policy mix provides a starting point to promote/maintain NBS, there is significant potential on the national level to uptake NBS into policy and optimize the rationale of budget allocation to design an optimized NBS network. Accordingly, we provide a wide strategic perspective that can support the allocation of funds currently envisaged by the EU Green Deal. We include the whole national territory in the identification, first of the challenges' distribution and then in the most effective and multifunctional NBS available for their mitigation.

## 3. Materials and Methods

This study was developed according to the three stages shown in Figure 1. Stage I: the identification and mapping of three environmental challenges in Italy (i.e., air quality, climate adaptation and mitigation, and water management), adopting the Environmental Quality Standards proposed by Di Pirro et al. [42]; Stage II: the overlay of the three challenges allows the identification of portions of territory threatened simultaneously by the same challenges (i.e., spatial groups); Stage III: a ranking of 24 NBS suitable to address the challenge(s) for each spatial group is proposed, based on the NBS performance assessment provided by Castellar et al. [30] and the land cover (Figure 1). All the analyses are conducted by a pixel-based approach.

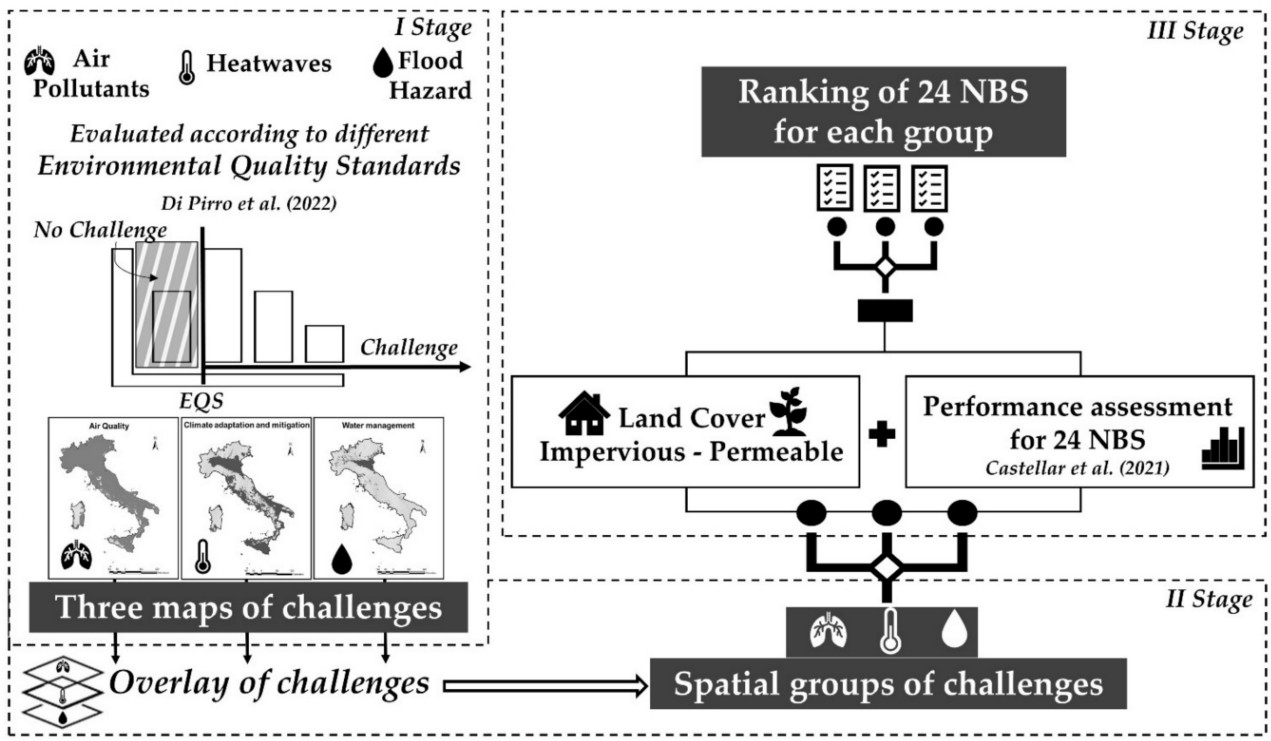

**Figure 1.** Workflow developed according to three main stages.

### 3.1. Environmental Challenges in Italy and Their Combination in Spatial Groups

According to Stage I (Figure 1), we considered three challenges, air quality, climate adaptation and mitigation, and water management (following Raymond et al. [15]), defined by the presence of three environmental stressors (air pollutants, frequency of heatwaves, flood hazard, respectively) altering human health when they exceed specific thresholds [61]

(e.g., Environmental Quality Standards—EQS [62]). Consequently, to identify the portion of national territory threatened by these challenges, we adopted the methodological framework proposed by Di Pirro et al. [42], where different EQS were selected and used as common thresholds to assess environmental and societal demands. The EQS proposed by Di Pirro et al. [42] are defined according to (i) the European standards set in Air Quality Directive (2008/50/EC), (ii) the definition of heatwaves and projections of climate change given by [43], and (iii) the flood hazard estimated by ISPRA [63] (further details are reported in [42]).

To define portions of the Italian territory showing air quality challenge, we considered EQS for the three most harmful pollutants in the EU, namely $PM_{10}$, $NO_2$, and $O_3$ [57]. All the pixels showing at least one of the three pollutants in exceedance for the respective EQS are considered as portions of territory where air quality regulation is needed to address the challenge. As regards the challenge of climate adaptation and mitigation, we considered the EQS of 4 days/year of heatwaves [43]. Hence, all the pixels exceeding this EQS are considered as portions of territory where climate regulation is needed to address the challenge. Finally, for the water management challenge, we considered EQS based on the probability that a flooding event will occur, according to estimates of their return period [63]. Hence, all the pixels in flood hazard are considered as portions of territory where water regulation is needed to address the challenge. Thus, starting from the methodological framework proposed by Di Pirro et al. [42], we derived three maps with the spatial distribution of the three challenges, with a spatial resolution of 1 km, as shown in Figure 2.

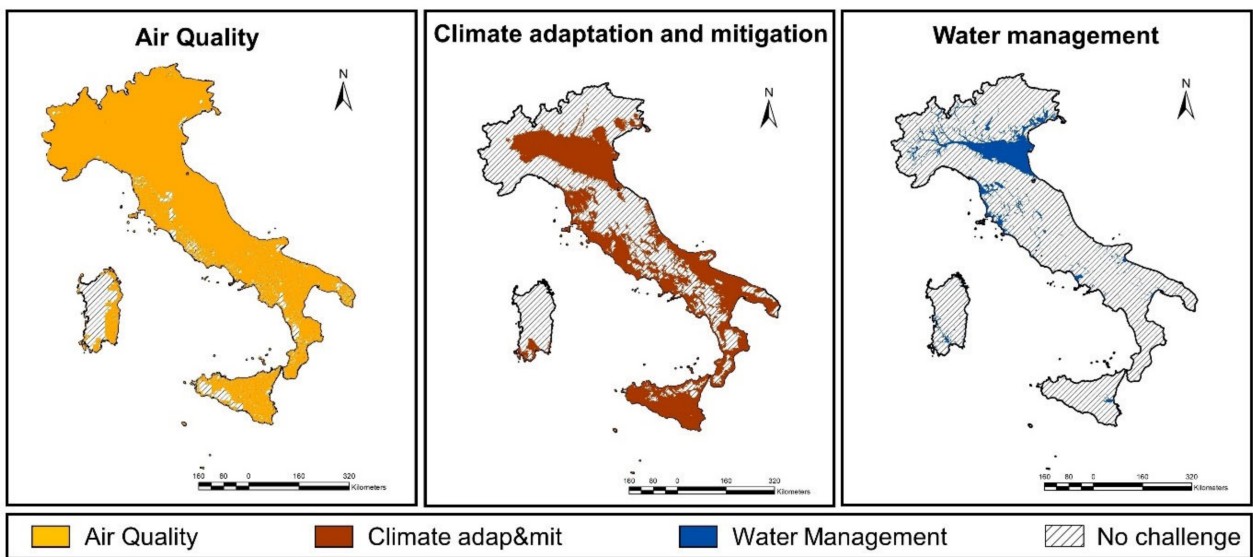

**Figure 2.** The three challenges considered: air quality, climate adaptation and mitigation, and water management. The areas with no challenge are represented by the striped pattern.

According to Stage II (Figure 1), the three maps of challenges were then combined in a GIS environment to investigate where, which, and how many challenges overlay in the same portion of the territory, thus needing to be addressed simultaneously. From this analysis, we obtained different homogenous groups, where interventions need to be differentiated to address the specific challenge(s). According to Stage III (Figure 1), for each group, we explored (i) the population density [64], to estimate the inhabitants exposed to single or multiple challenges as well as the potential beneficiaries of NBS; and (ii) the land cover, to define quantity and typology of space available for NBS implementation. We focused our analysis on two land covers (i.e., impervious and permeable) using the 2018 High-Resolution Layers, with a spatial resolution of 20 m [65]. The impervious surfaces were reclassified according to Congedo et al. [66], thus considering values of the Degree of

Imperviousness greater than 29%. The permanent water bodies (covering about 1.4% of the national territory) were neglected, since specific policies (e.g., water quality and security) and NBS might be implemented, out of the scope of this work.

### *3.2. Calculating the Nature Based Solutions Performance in Dealing with Challenges*

Relying on the NBS capacity to provide multiple ES and mitigate environmental stressors, we assumed that NBS are the unique interventions to address the challenges in each group. In the literature, NBS encompasses a wide range of interventions and actions. Following the classification proposed by Eggermont et al. [67], NBS can be considered as conservation and restoration of ecosystems (i.e., Type 1), sustainable management for improving ES supply (i.e., Type 2), and the creation of new ecosystems (i.e., Type 3). For this work, we considered only the creation of new ecosystems, i.e., NBS Type 3 according to Eggermont et al. [67], focusing in particular on NBS spatial and technological units proposed by Castellar et al. [30]. New NBS thus need to be identified and differentiated according to their capacity to provide ES able to address the challenge(s) (i.e., performance). We adopted the performance assessment proposed by Castellar et al. [30], where for 32 NBS they calculated a performance score (PS), ranging from 0 to 1, representing the NBS capacity to provide ES able to address ten different challenges. We limited our analysis to 24 NBS and respective PS related to the three challenges under investigation in this study (i.e., air quality, climate adaptation and mitigation, and water management). Eight NBS were thus excluded since they are not terrestrial or not considered as Type 3 (i.e., new ecosystem). Therefore, when a single challenge characterizes the group, we reported the same PS of Castellar et al. [30]; when multiple challenges characterize the group, we averaged the PS for the respective challenges of the group. Accordingly, starting from the 24 NBS, we produced different rankings of PS as many as the groups of challenges, thus allowing us to select the best performing set of NBS to effectively address the environmental challenges.

### *3.3. Classification of Nature Based Solutions for Land Covers*

The 24 NBS considered for this study are the following: community gardens, constructed wetlands, extensive green roofs, green corridors, green façades, green wall systems, heritage gardens, infiltration basins, intensive green roofs, large urban parks, planter green walls, pocket gardens/parks, private gardens, raingardens, semi-intensive green roofs, street trees, swales, urban forests, urban orchards, vegetated grid paves, vegetated pergolas, vertical mobile gardens, (wet) retention ponds, and shelters for biodiversity.

We classified NBS into two categories: implementable in impervious surfaces (I-NBS) and permeable surfaces (P-NBS). The classification is based on descriptions provided by Castellar et al. [30], considering both the NBS' structures and sizes. Accordingly, I-NBS are those implementable on buildings (i.e., green façades, green wall systems, vertical mobile gardens, planter green walls, vegetated pergolas, extensive green roofs, intensive green roofs, semi-intensive green roofs), and along streets and parking lots, close to buildings and houses (i.e., raingardens, swales, street trees, and vegetated grid paves, pocket gardens/parks, private gardens). Except for vegetated grid paves, where we consider the conversion from traditional car parks to green parking lots, we excluded the possibility of de-sealing actions (land cover changes from impervious to permeable), e.g., building's removal and conversion to a large urban park. On the other side, P-NBS are those that can be implemented on permeable land covers (green corridors, large urban parks, urban forests, heritage gardens, community gardens, urban orchards, infiltration basins, (wet) retention ponds, constructed wetlands). NBS similar to each other for the structure but not for the size (e.g., pocket gardens vs large parks) were distinguished by a 0.5 ha threshold [30]. Consistently with the HRL spatial resolution used to estimate land covers surfaces [65], we reduced this dimensional threshold to 400 m$^2$ (i.e., one pixel). In this way, pocket gardens were assigned to I-NBS while large parks were assigned to P-NBS.

Therefore, for each group, we can provide the quantity and typology of challenges to address, the incidence of permeable and impervious land covers, a ranking of P-NBS

and I-NBS ranging from 0 to 1 based on their ability to address the specific challenges of the group.

## 4. Results

Eight different spatial groups resulted from the spatial combination of the three challenges; Figure 3 shows the map with their spatial distribution while the pie-chart shows the relative coverage of each spatial group. Only 6% of the national territory shows no challenge ("NoChal" group). Conversely, 7.9% of the territory shows all the three challenges combined simultaneously ("ALL" group). Three groups show the individual challenge covering 51.5% of the national territory: 47% air quality ("AIR" group), 4.3% climate adaptation and mitigation ("CLIM" group), and 0.2% water management flood hazard ("WAT" group). The remaining 35% of the territory is occupied by the last three groups, characterized by two challenges simultaneously. The challenge of air quality co-occurring with climate adaptation and mitigation covers 33% of the national territory ("AIR-CLIM" group), while its combination to water management spans over 1.2% of the national territory ("AIR-WAT" group). Finally, when the spatial combination is between climate adaptation and mitigation and water management, the group covers 0.3% of the country ("CLIM-WAT" group).

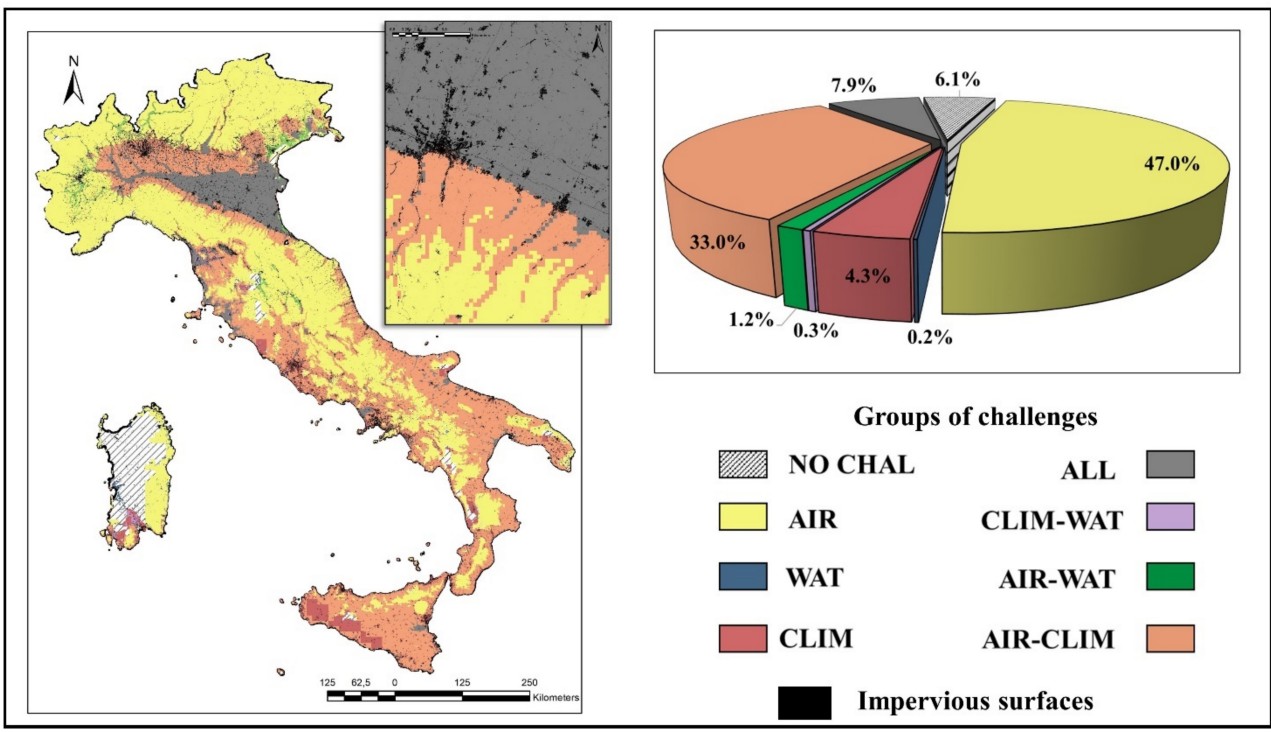

**Figure 3.** The eight spatial groups of challenges. The pie chart shows the relative coverage of each spatial group (% of the national territory). In black on the map is shown the spatial distribution of impervious surfaces throughout the national territory.

Built-up areas in Italy cover about 7.1% of the national territory [47], and their incidence is quite variable across the groups (Table 1 and Figure 3). The AIR and the NoChal groups are the only ones showing a relatively impervious surface lower than the national one (3% and 4%) as well as the lowest population density (91 and 150 inhab/km$^2$). On the contrary, CLIM and CLIM-WAT groups show the highest relative impervious surfaces (18% and 24% respectively) as well as the highest population density (1036 and 1086 inhab/km$^2$). AIR-WAT, ALL, WAT, and AIR-CLIM groups, respectively, show the following relative impervious surfaces, slightly higher than the national one: 13%, 11%, 9%, and 8%, with intermediate values of population density, 326, 257, 254, and 213 inhab/km$^2$.

**Table 1.** For each group, the area in km$^2$, coverage of permeable and impervious surface (km2), number of inhabitants, and population density (inhab/km2) within the groups are reported.

| Groups | Area (km$^2$) | Permeable (km$^2$) | Impervious (km$^2$) | Population (n° inhab) | Pop Dens (inhab/km$^2$) |
|---|---|---|---|---|---|
| AIR | 141,044 | 136,310 | 4734 | 12,974,163 | 91 |
| CLIM | 12,582 | 10,345 | 2237 | 13,468,447 | 1036 |
| WAT | 477 | 435 | 42 | 130,275 | 254 |
| AIR-CLIM | 97,769 | 89,603 | 8166 | 21,377,514 | 213 |
| AIR-WAT | 3352 | 2904 | 448 | 1,238,194 | 326 |
| CLIM-WAT | 1043 | 790 | 253 | 1,152,988 | 1086 |
| ALL | 22,875 | 20,446 | 2429 | 6,163,604 | 258 |
| NoChal | 18,393 | 17,639 | 754 | 2,802,590 | 150 |

All the 24 NBS show Performance Scores (PS) ranging from the minimum value of 0, only in the groups characterized by single challenges, to the maximum of 1, in each group. We divided PS into three classes, low PS (0–0.33), medium PS (0.33–0.66), high PS (0.66–1), to more facilitate the reading of the different performances.

The number of NBS with high PS (Table 2) is variable across the groups ranging from 16 in the WAT group to 11 in AIR-WAT. NBS with high PS can be implemented in both permeable (with a maximum of 9 P-NBS in the WAT group) and impervious land covers (with a maximum of 9 I-NBS in the CLIM group). P-NBS have similar PS and ranking among the different groups, while we registered more dissimilarity among the I-NBS both for PS values and ranking. This difference is particularly marked for I-NBS in AIR and WAT groups. Accordingly, their combination in the AIR-WAT group shows the least number of high PS (4 I-NBS), highlighting a lack of synergy between ES for simultaneously addressing the challenges of air quality and water management.

Vertical green (i.e., green façades, green wall systems, vegetated pergola, vertical mobile gardens) shows high PS for the mitigation of both air pollutants and heatwaves while low PS for the mitigation of flood hazard. We found the opposite PS trend for rain gardens, swales, and vegetated grid paves, which are particularly useful to mitigate flood hazards and so address the challenge of water management.

Seven NBS have high PS simultaneously in all groups: five P-NBS (i.e., infiltration basin, green corridors, urban forests, large urban park, heritage garden), and two I-NBS (i.e., intensive green roof, semi-intensive green roof). Hence, thanks to these co-benefits, these seven NBS can be potentially implemented throughout 94% of the Italian territory, thus ensuring good performances employing less than one-third of the available NBS. On the contrary, among P-NBS, urban orchards and planter green walls show the lowest PS in all groups, thus representing a sub-optimal solution for addressing the three environmental challenges considered here (Table 2).

Figure 4 shows a specific focus on the distribution of impervious land cover within the spatial groups. Among the 20,000 km$^2$ of built-up areas in Italy, 8100 km$^2$ are occupied by the AIR-CLIM group (42.8%), 4700 km$^2$ by the AIR group (24.8%), over 2200 km$^2$ by the ALL and CLIM groups (12.7 and 11.7%, respectively), about 750 km$^2$ by the NoChal group (4%), and less than 450 km$^2$ by the AIR-WAT, CLIM-WAT, and WAT groups. Therefore, combining these coverages with the NBS having high-PS in each spatial group, we obtained the overall area where both P-NBS and I-NBS could potentially be implemented to address multiple challenges. With specific regard to the I-NBS: intensive and semi-intensive green roof (18,309 km$^2$), street trees (17,819 km$^2$), green façade (17,566 km$^2$), green wall system and vertical mobile garden (15,137 km$^2$), private gardens (13,127 km$^2$), and extensive green roof (13,085 km$^2$) (Figure 4).

**Table 2.** Twenty-four Nature-Based Solutions (NBS) classified as implementable in impervious (I-NBS) and permeable (P-NSB) land covers. Performance Score (PS) ranges between 0 (low performance) and 1 (high performance) for each NBS in the 7 groups, namely: AIR, CLIM, WAT, AIR-WAT, AIR-CLIM, CLIM-WAT, ALL. The black triangles mark NBS with high-PS (PS > 0.66). The NoChal group is not included considering we assumed that no new NBS are needed.

| Nature Based Solutions | Performance Score (PS) | | | | | | |
|---|---|---|---|---|---|---|---|
| **I-NBS** | **AIR** | **CLIM** | **WAT** | **AIR-CLIM** | **AIR-WAT** | **CLIM-WAT** | **ALL** |
| Extensive green roofs | 0.5 | ▲0.9 | 0.6 | ▲0.7 | 0.5 | ▲0.7 | ▲0.7 |
| Green walls system | ▲1.0 | ▲0.8 | 0.0 | ▲0.9 | 0.5 | 0.4 | 0.6 |
| Green façades | ▲1.0 | ▲1.0 | 0.2 | ▲1.0 | 0.6 | 0.6 | ▲0.7 |
| Intensive green roofs | 0.7 | ▲0.9 | ▲0.8 | ▲0.8 | ▲0.8 | ▲0.9 | ▲0.8 |
| Planter green walls | 0.5 | 0.5 | 0.0 | 0.5 | 0.3 | 0.3 | 0.3 |
| Pocket gardens/parks | 0.6 | 0.6 | ▲0.8 | 0.6 | ▲0.7 | ▲0.7 | ▲0.7 |
| Private gardens | 0.5 | ▲1.0 | ▲0.8 | ▲0.8 | 0.6 | ▲0.9 | ▲0.8 |
| Raingardens | 0.4 | 0.3 | ▲0.8 | 0.4 | 0.6 | 0.6 | 0.5 |
| Semi-intensive green roofs | ▲0.7 | ▲0.8 | ▲1.0 | ▲0.8 | ▲0.8 | ▲0.9 | ▲0.8 |
| Street trees | ▲0.8 | ▲0.9 | 0.4 | ▲0.8 | 0.6 | ▲0.7 | ▲0.7 |
| Swales | 0.6 | 0.2 | ▲0.9 | 0.4 | ▲0.7 | 0.5 | 0.6 |
| Vegetated grid paves | 0.2 | 0.5 | ▲0.8 | 0.3 | 0.5 | 0.6 | 0.5 |
| Vegetated pergola | 0.5 | ▲0.8 | 0.3 | 0.6 | 0.4 | 0.5 | 0.5 |
| Vertical mobile garden | ▲1.0 | ▲0.9 | 0.0 | ▲1.0 | 0.5 | 0.5 | 0.6 |
| **P-NBS** | | | | | | | |
| (Wet)Retention Ponds | ▲0.8 | 0.6 | ▲1.0 | ▲0.7 | ▲0.9 | ▲0.8 | ▲0.8 |
| Community gardens | 0.3 | 0.5 | ▲0.8 | 0.4 | 0.6 | ▲0.7 | 0.6 |
| Constructed wetlands | 0.0 | 0.3 | ▲1.0 | 0.1 | 0.5 | 0.6 | 0.4 |
| Green Corridors | ▲1.0 | ▲1.0 | ▲0.7 | ▲1.0 | ▲0.8 | ▲0.8 | ▲0.9 |
| Heritage gardens | ▲1.0 | ▲1.0 | ▲1.0 | ▲1.0 | ▲1.0 | ▲1.0 | ▲1.0 |
| Infiltration basins | ▲0.8 | ▲0.8 | ▲1.0 | ▲0.8 | ▲0.9 | ▲0.9 | ▲0.9 |
| Large urban parks | ▲1.0 | ▲1.0 | ▲0.9 | ▲1.0 | ▲1.0 | ▲1.0 | ▲1.0 |
| Shelters for biodiversity | ▲0.8 | 0.0 | ▲0.7 | 0.4 | ▲0.7 | 0.3 | 0.5 |
| Urban forests | ▲1.0 | ▲0.9 | ▲0.8 | ▲0.9 | ▲0.9 | ▲0.9 | ▲0.9 |
| Urban orchards | 0.3 | 0.2 | 0.5 | 0.3 | 0.4 | 0.3 | 0.3 |

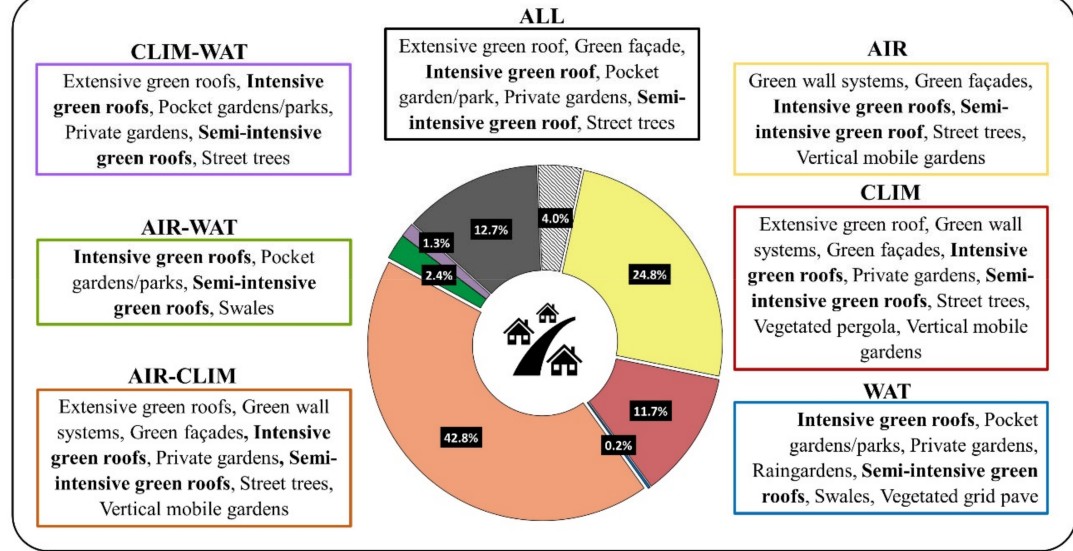

**Figure 4.** The pie chart shows the distribution of the spatial groups within the total impervious surface in Italy; the striped pattern represents the NoChal group. The I-NBS with high PS are reported in correspondence of each spatial group, with the I- NBS showing high PS simultaneously in all spatial groups marked in bold.

## 5. Discussion

Our framework has implications for the future development of cross-scale strategies to reach multiple national targets through NBS. It highlights the need to consider the multiple challenges to tackle as a key criterion to improve the NBS co-benefits and cost-effectiveness. Current NBS (or related concept) planning frameworks usually tend to focus on a single ES supply or address a specific challenge [41,68] and with a specific focus on the municipality scale [69–71]. However, our results highlighted that about 42% of the national territory shows multiple challenges simultaneously, and 50% of the population is exposed to these critical conditions. In these spatial groups (AIR-CLIM, AIR-WAT, CLIM-WAT, and ALL), multiple targets need to be achieved, through implementing NBS with the best performance to provide multiple ES. Consequently, the widespread distribution of areas under multiple challenges underlines that (i) in the political agenda, actions for air quality improvement might be coupled with those of climate change mitigation and adaptation [15]; (ii) considering both multiple ES demand (i.e., challenges to address) and multiple ES supply (i.e., NBS performance) has a key role to maximize the cost-effectiveness of interventions and the optimal use of the available space [8]. Europe—and the Member States—need to effectively leverage investments in NBS provided by the Green Deal, developing strategies to generate gains for adaptation, mitigation, disaster risk reduction, biodiversity, and health [37]. Therefore, the definition of a national intervention priority based on the intensity of challenges and population exposed [42] combined with the NBS performance ranking provided by our framework could help in this path, optimizing investment allocation from the national to the local governments.

Of the 24 NBS assessed, all spatial groups show from 11 to 16 NBS with high PS, both on impervious and permeable land covers. Providing a defined set of NBS is crucial for decision-makers and planners given the variety of NBS available [72], with different nomenclature, as well as the numerous barriers that may arise in urban areas from the planning stage to the site-scale design and implementation. The 24 NBS we considered in this work were selected from Castellar et al. [30], where, through using different workshops and surveys, they evaluated their performance to meet ten challenges, including the supply of all categories of ES. In the present work, some NBS may show similarities or overlapping results, being limited to only regulation ES (i.e., mitigation of air pollutants, heatwaves, and flood hazards). This could stand as a limitation; however, we decided not to further manipulate the nomenclature, thus leaving the possibility to replicate our methodology by also including other ES (e.g., provisioning, cultural, supporting) and other challenges (e.g., social cohesion).

Furthermore, the surfaces we evaluated as potentially available for the implementation of high-performing NBS do not necessarily correspond with the real space availability. Due to the broad scale and the aim of the work, we did not consider, e.g., archaeological constraints, protected areas, limited space in historical centers, that could decrease the suitability and space availability for NBS implementation. Therefore, for the local-scale implementation of the NBS, an in-depth assessment is necessary to include many other biophysical, economic, and social variables. To conduct a more detailed analysis and support the local scale governance to overcome the over-mentioned barriers, other layers could be useful, e.g., implementation and maintenance costs, the urban form, endemic vegetation, the public opinion, and many others that would be out of the scope and scale of this study.

However, according to the aim of this work, the incidence of land covers (i.e., impervious and permeable) in each spatial group, combined with the population density, suggests (i) which combination of factors is most related to the built-up areas, (ii) which risks the population is mainly exposed to, (iii) where to localize the interventions, and (iv) the number of beneficiaries of the expected increase in ES supply. On the one hand, we found spatial groups showing both high incidences of impervious surfaces and high population density, where I-NBS might be preferred. On the other hand, we found spatial groups with a lower incidence of permeable surfaces and low population density, where

widespread and large-scale P-NBS (e.g., large urban parks, urban forests) in the territorial matrix could be more adequate. For example, impervious surfaces in the CLIM-WAT group have a built-up area's incidence eight times higher than in the AIR group (24% vs. 3%) and a population density twelve times greater too. This result suggests that investing equal resources (e.g., budget) in the first group, I-NBS, would affect more beneficiaries in a smaller area, mitigating both heatwaves and flood hazards, hence addressing two challenges simultaneously. These findings are particularly helpful since limited available space can act as barriers to NBS implementation, especially in urban areas where land is a scarce and expensive resource [21]. Potentially, some I-NBS (i.e., vertical green), even if less performing than others in P-NBS, have the advantage to occupy space often unemployed [73] and consequently not contributing to exacerbating conflicts around open space (e.g., land use change) in densely built-up areas [74].

### 5.1. Nature Based Solutions Implementable in Impervious Land Cover

Our results show that intensive and semi-intensive green roofs can potentially be implemented on 18,300 km$^2$, showing high PS in all spatial groups and hence standing as the most versatile and effective NBS among all the I-NBS assessed in this work. Although intensive and semi-intensive green roofs were initially designed for water management [75], due to their more deeply planted vegetation [73], they are also proved to positively contribute to climate mitigation, air quality, and biodiversity.

In terms of coverage and performance, among I-NBS, street trees and private gardens have high PS in five spatial groups and can be implemented, respectively, across 17,800 km$^2$ and 13,100 km$^2$, usually close to buildings, houses, and streets. Street trees show the best performance for the mitigation of air pollutants (AIR) and heatwaves (CLIM), both individually and combined (AIR-CLIM). Furthermore, the species selection can help both to mitigate pollutants [76,77] and to provide shade by the crown coverage [78,79]. Otherwise, when heatwaves and flood hazards need to be simultaneously mitigated, private gardens are more effective than street trees, contributing both to water management through the broadest unsealed soils, and to enhance air circulation and cooling through plant transpiration and shading [80]. Similarly, vertical green solutions (i.e., green façades, vertical mobile gardens, and green wall systems), are mainly reliable to stock air pollutants [81] and heatwave mitigation [82,83]. These I-NBS perform better in AIR, CLIM, and AIR-CLIM groups, covering potentially 17,500 km$^2$ of impervious surface. On the other hand, extensive green roofs show high PS for heatwaves and flood hazards [84], hence represented within AIR-CLIM, CLIM-WAT, CLIM, and ALL groups, and covering 13,000 km$^2$ of impervious surface.

### 5.2. Nature Based Solutions Implementable in Permeable Land Cover

Unlike the built-up areas where NBS Type 3 are usually considered, before implementing P-NBS, it is first necessary to evaluate the current land uses to consider their conservation (Type 1) and management (Type 2) instead of the substitution with new ecosystems (Type 3). This is following what was observed by Sarabi et al. [21], i.e., the entity of interventions required in NBS increases when moving (closer) to the center of built-up contexts, and vice versa. Therefore, in the case of currently forested areas, the objective should focus on their preservation, restoration, or enhancement to maximize ES supply (Type 1 and Type 2 [67]). This is a relevant option, for example, in the case of the AIR group, mainly occupied by forested areas. This is in line with the trajectory identified by the EU Green Deal, where, along with the protection and management of existing forests, urban, peri-urban, and agricultural areas need to be integrated with additional trees (i.e., 3 billion trees [23]). Accordingly, the assessed P-NBS can be applied in marginal, abandoned, unproductive, peri-urban areas [85–87], since they are considered as new ecosystems (Type 3 [67]).

Despite the finding that five P-NBS have high PS in all spatial groups (green corridors, large urban parks, heritage gardens, infiltration basins, and urban forests), they are similar

to each other for regulating ES supply and thus addressing the challenges we considered in this work. The choice to implement one P-NBS instead of others can be related to the supply of other ES categories (cultural, provisioning, and support) as well as to other policy and planning issues (e.g., people perceptions, recreation needs) and budgetary constraints. Particularly, large urban parks and heritage gardens refer to large green areas (>0.5 ha) with mixed land uses (i.e., forests, grasslands, ponds). The first ones are mainly oriented to provide a variety of recreational facilities, mainly addressing the social demands of the residents, while the second ones aim to preserve outstanding historical, cultural, aesthetic, or scientific value [30]. Moreover, co-benefits and multifunctionality (i.e., multiple ES supply) of the single NBS can be enhanced by adding some improvements, usually including tree planting. As an example, infiltration basins can be partially forested to fulfill other functions such as providing recreational spaces for inhabitants, increasing biodiversity and ecological connectivity [88]. Similarly, green corridors are usually renatured areas of derelict infrastructure (i.e., railway) or placed along rivers. They can be afforested where there is the need to enhance landscape connectivity and ecological restoration [70,89,90]. Furthermore, their social role can be emphasized by ameliorating the availability and accessibility of currently vacant and underused land in urban contexts [91]. Therefore, the five P-NBS considered here already include—or could include—individual trees and/or groups of trees, as they are considered to be the best natural elements to increase the spectrum of ES provided [26,79,85,92–94].

## 6. Conclusions

The environmental challenges addressed in this work can adversely affect human health and well-being, with associated mitigation costs. Accordingly, our work contains a novel framework that will help both the national government and the municipalities to identify NBS able to maximize the ES supply while addressing multiple challenges. Analogously to the already proposed "National Strategy on Urban Greenspaces" [59], this work can provide a strategic vision at the national scale, but it can be consulted and adopted by all municipalities as a common roadmap, also helpful in facing the recurring problematic of planning silos. Indeed, the relevance of our framework is not just focused on the NBS application at the local scale but also shows a great impact on a wider scale (e.g., national and regional).

On a national scale, the framework proposed here can reliably (i) identify the areas showing a simultaneous demand for the achievement of multiple national targets; (ii) spatially orient the new investment needed to mitigate the challenges (e.g., EU Green Deal); and (iii) support the NBS selection that provides more co-benefits, playing a crucial role in increasing budget allocations efficiency.

On a municipal scale, the NBS ranking can be used as a guideline for further specific planning and design activities based on local issues, barriers, and peculiarities, while remaining consistent with national targets.

Italy is currently allocating funds in the 14 metropolitan cities to implement urban forests. Our results confirm that urban forests are among the best performing NBS, and Di Pirro et al. [42] argue that reforestation programs could also be expanded to other municipalities with few additional resources (+7.5% of the national territory) but involving an extra 46% of the national population. Although trees and forests (especially urban ones) are considered by many authors as the best solution to address environmental challenges [79,85,92,93], our work also proposes a list of performing I-NBS (e.g., green roofs) that can be implemented on sealed surfaces. These can help mitigate environmental stressors by using impervious surfaces i) that are usually unemployed (e.g., gravel or bitumen roofs) and ii) that could even exacerbate the challenges due to their physical characteristics (e.g., thermal emittance, reduced infiltration capacity) [95]. This option can also contribute to mitigating the negative effects related to soil sealing, which is a remarkable issue in the EU [96,97], enhancing the values of interstitial and leftover spaces [87]. However, the technical feasibility and costs related to these I-NBS and their widespread implementation

must be evaluated according to the specific local conditions [73]. Finally, at the local scale, additional co-benefits (i.e., energy savings, biodiversity increase, and social cohesion), as well as possible disservices (i.e., BVOC emissions, decrease in wind velocity, gentrification), should also be included for a more overarching assessment [94,98,99]. Planning and managing NBS can be approached holistically [40], considering diverse benefits concerning different spatial–temporal scales. The multi-scale approach can help in considering different stakeholders as well as social, economic, and biophysical characteristics that matter in the benefit provision and are thus better included in decision-making related to national, regional, city/site-scale spatial plans [100].

**Author Contributions:** Conceptualization, E.D.P. and L.S.; methodology, E.D.P., L.S. and J.A.C.C.; software, formal analysis and data curation, E.D.P.; validation, E.D.P., L.S., J.A.C.C., G.S. and B.L.; investigation, E.D.P., L.S. and G.S.; resources, M.M. and B.L.; writing—original draft preparation, E.D.P.; writing—review and editing, E.D.P., L.S., J.A.C.C., G.S., M.M. and B.L.; visualization, E.D.P., G.S. and M.M.; supervision, M.M. and B.L.; project administration and funding acquisition, B.L. All authors have read and agreed to the published version of the manuscript.

**Funding:** This work has received funding from the research project "Establishing Urban FORest based solutions In Changing Cities" (EUFORICC), cod 20173RRN2S, funded by the PRIN 2017 program of the Italian Ministry of University and Research (project coordinator: C. Calfapietra).

**Institutional Review Board Statement:** Not applicable.

**Informed Consent Statement:** Not applicable.

**Data Availability Statement:** The data presented in this study are available on request from the corresponding author.

**Conflicts of Interest:** The authors declare no conflict of interest.

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
