# Peer review of "Facing Multiple Environmental Challenges through Maximizing the Co-Benefits of Nature-Based Solutions at a National Scale in Italy"

_forests, doi:10.3390/f13040548_

Round 1

Reviewer 1 Report

General comments

The authors capitalize several terms that do not seem to need capitalization, including nature-based solutions, ecosystem services, climate change, urban forests, green corridors, and other solutions. Only proper nouns and names of formal programs should be capitalized.

The vague term “groups” in the abstract confused me, but as I read the methods, I understood better. Fig 1 effectively lays out the authors’ approach. To make things clearer for readings, I suggest using “spatial clusters” or “spatial groups” everywhere that you currently say simply “groups”.

The authors need a robust limitations paragraph in their discussion. Address the limitations of the modeling approach itself. Models of this kind are always imperfect representations of reality. The analysis seems reasonable but please help guide readers through shortcomings. What additional data might improve the models? What could help make the models most useful to policy-makers?

I appreciate the mention of disservices in the conclusion section as a topic for further research.

Here is my biggest concern with this paper: In section 3.3, the NBS need to be defined / described in terms of how the authors used them in the study. By my understanding of these terms, many of the 24 listed NBS overlap, which muddles the analysis considerably. In other words, these NBS are not mutually exclusive. For example, if by “urban forest” the authors mean all trees in urbanized areas, then many of the other NBS categories include urban forests (such as street trees, community gardens, urban orchards, urban parks, private gardens – any tree elements of these NBS categories could be considered part of the urban forest). I see that the authors cite other publications regarding this categorization of NBS. Many of these NBS terms are used and defined in different ways in the literature and in practice, and there can be contested meanings, so clarity in definitions within this manuscript is essential.

Specific editing suggestions

L14: minor edit for readability, “… in the Green Deal, including introducing…”

L67: thank you for raising the challenge of planning silos. This is a major roadblock to effective implementation of NBS.

L90: extremely awkward and unclear wording, perhaps you mean “Despite the fact that NBS are ….”

Author Response

#1.1 The authors capitalize several terms that do not seem to need capitalization, including nature-based solutions, ecosystem services, climate change, urban forests, green corridors, and other solutions. Only proper nouns and names of formal programs should be capitalized.

Rep: Thank you for reporting this inaccuracy. We left capital letters for the terms which the acronyms have been then reported (e.g., Nature-Based Solutions-NBS) or the names of programs, laws, strategies (e.g., Green Deal, Horizon2020, National Strategy of Urban Green Spaces). As you suggested, we report in lowercase all the names of the NBS. 

#1.2 The vague term “groups” in the abstract confused me, but as I read the methods, I understood better. Fig 1 effectively lays out the authors’ approach. To make things clearer for readings, I suggest using “spatial clusters” or “spatial groups” everywhere that you currently say simply “groups”.

Rep: Thanks for this suggestion. Indeed, the term "groups" is very vague and it can confuse the reader. We changed throughout the text from "groups" to "spatial groups" trying to be clearer and more specific to the readers. We chose not to use "spatial cluster"  to avoid any kind of inaccuracy since we did not conduct statistical analysis regarding the spatial overlap of the challenges. In the abstract, due to characters limitations, we reported only for the first time that appears in the text the term "spatial group". 

#1.3 The authors need a robust limitations paragraph in their discussion. Address the limitations of the modeling approach itself. Models of this kind are always imperfect representations of reality. The analysis seems reasonable but please help guide readers through shortcomings. What additional data might improve the models? What could help make the models most useful to policy-makers?

Rep: Thank you for highlighting this. In the previous version of the manuscript, we indeed missed strictly focusing on the limitations aspect. Accordingly, we now specified in the discussion section why we chose to maintain all the 24 NBS despite the apparent nomenclature overlap, as well as the limitation related to the available and suitable surfaces for NBS implementation. The modified version stands as follows "The 24 NBS we considered in this work were selected from Castellar et al [30] where, through different workshops and surveys, they evaluated their performance to meet ten challenges, including the supply of all categories of ES. In the present work, some NBS may show similarities or overlapping results being limited to only regulation ES (i.e., mitigation of air pollutants, heatwaves, and flood hazards). This could stand as a limitation, however, we decided not to further manipulate the nomenclature, thus leaving the possibility to extend our methodology to other ES (e.g., provisioning, cultural, supporting) and challenges (e.g., social cohesion). Furthermore, the surfaces we evaluated as potentially available for the implementation of high-performing NBS not necessarily correspond with the real space availability.  Due to the large scale and the aim of the work, we did not consider, e.g., archaeological constraints, protected areas, limited space in historical centers, that could decrease the suitability and space availability for NBS implementation. Therefore, for the local-scale implementation of the NBS, an in-depth assessment is necessary to include other biophysical, economic, and social variables. To conduct a more detailed analysis and support the local scale governance to overcome the over-mentioned barriers, other layers could be useful, e.g., implementation and maintenance costs, the urban form, endemic vegetation, the public opinion, and many others that would be out of the scope (and the scale) of this study” (lines 375-394). With particular regard to the P-NBS, this aspect is also underlined in paragraph 5.2, lines 452-458. The intervention feasibility and further data needed for a local scale planning are also reported in the Concluding Remarks section, as follows (line 506) "However, the technical feasibility and costs related to these I-NBS and their widespread implementation must be evaluated according to the specific local conditions [73]. Finally, at the local scale, additional co-benefits (i.e., energy savings, biodiversity increase, social cohesion), as well as possible disservices (i.e., BVOC emissions, decrease in wind velocity, gentrification), should also be included for a more overarching assessment [94,98,99]."

#1.4 I appreciate the mention of disservices in the conclusion section as a topic for further research.

Rep: Thank you, we really appreciate that you noticed this. We agree, the issue of disservices is often neglected and should be further considered and explored.

 #1.5 Here is my biggest concern with this paper: In section 3.3, the NBS need to be defined/described in terms of how the authors used them in the study. By my understanding of these terms, many of the 24 listed NBS overlap, which muddles the analysis considerably. In other words, these NBS are not mutually exclusive. For example, if by “urban forest” the authors mean all trees in urbanized areas, then many of the other NBS categories include urban forests (such as street trees, community gardens, urban orchards, urban parks, private gardens – any tree elements of these NBS categories could be considered part of the urban forest). I see that the authors cite other publications regarding this categorization of NBS. Many of these NBS terms are used and defined in different ways in the literature and in practice, and there can be contested meanings, so clarity in definitions within this manuscript is essential.

Rep: Thank you for highlighting this point. The 24 NBS we considered refer to the work of Castellar et al., 2021. Through surveys and workshops, they considered the ability of the NBS to meet ten urban challenges and supply of ecosystem services. Their work focused precisely on finding a common nomenclature and avoiding any kind of overlap in their definition. Therefore the 24 NBS we used show consistent differences from each other and they thus are mutually exclusive. Being this nomenclature aspect little bit out of the specific aim of this work, for specific and detailed information regarding their methodology we suggest consulting the paper by Castellar et al., 2021. The limitation in our application lies in the fact that we only considered the performance of regulation ecosystem services related to three environmental challenges. Therefore, the overlap of some results is attributable to this problem. We chose not to further modify the nomenclature since we consider it more appropriate to i) adopt a common nomenclature that is already reliable and published, and ii) make our approach replicable and extendable to other ecosystem services as well as other urban challenges. We reported this limitation in two different points in the discussions section, lines 371-383 and lines 452-473. 

#1.6 Specific editing suggestions 

L14: minor edit for readability, “… in the Green Deal, including introducing…”

Rep: Done.

 L67: thank you for raising the challenge of planning silos. This is a major roadblock to effective implementation of NBS.

Rep: thank you. Regarding the planning silos, we hope that the methodology proposed in our work could help to face this challenge. Accordingly, we report the concept in the Concluding Remarks section on line 478, as follows “this work can provide a strategic vision at the national scale but it can be consulted and adopted by all municipalities as a common roadmap, also helpful in facing the recurring problematic of planning silos.” 

L90: extremely awkward and unclear wording, perhaps you mean “Despite the fact that NBS are ….”

Rep: We correct according to your suggestions.

Reviewer 2 Report

The issue addressed in the paper discusses the maximizing the co-benefits of Nature-Based Solutions at a national scale in Italy. Indeed, these are important environmental challenges that the Green Deal should support. But is the overall case study for the Italy area as a whole really precisely defined? Perhaps more detail is needed in the spatial scope of the study?

First of all, I find that an important topic, compatible with the journal's scope, was considered.

Such studies are partially analysed in literature. It would be worth presenting the state of the art in a broader way. I suggest a more dilligent, comparative description of other scientific research from the literature (for example, it is possible to add a short state of the art comparative analysis report).

I also recommend several corrections to improve the quality of this paper:

- to precisely define the research scenario (it is very general); needed to clarify the scope of the study and consequently a clear, step-by-step, simple, synthetic research pattern; yes, the methodology is described, but I recommend more precision, as the reader should know how to repeat a similar analysis on this basis (please consistently correct and complete section 3);

- to improve the readability and description of figures (since they are the basis for analysis verification), supplement the history of their description, a clear and not laconic reference in the paper;

that is, supplement the discussion and summary descriptive analysis (please complete section 5) .

Please remember that the formulated objectives - find a clear answer in the conclusion of the study. Is this really the way it works?

Does the conclusion answer all the questions posed at the beginning of the paper (expressed in objectives and hypotheses)? Please complete it and also correct it. The conclusion needs to be supplemented (section 6). I also strongly suggest that recommendations for specific, practical, not only general (and not entirely clear) applications of this research shall be provided.

The language of this paper is relatively correct, however some descriptions would benefit from being more concise.

Author Response

#2.1 The issue addressed in the paper discusses the maximizing the co-benefits of Nature-Based Solutions at a national scale in Italy. Indeed, these are important environmental challenges that the Green Deal should support. But is the overall case study for the Italy area as a whole really precisely defined? Perhaps more detail is needed in the spatial scope of the study?

Rep: Thanks for this comment. We believe that Italy is a suitable case study because i) it is a member state receiving funds from the European Union to mitigate environmental challenges and implement NBS, ii) it is one of the countries that present the most incidence of the challenges examined in this work (effects adverse to climate change and air pollution), iii) in addition to the EU budget, the Italian national government is currently investing and allocating funds to some administrative areas following only the criterion of population density. All these points are thoroughly explained in paragraph 2 "case study". However, according to your suggestion, we reported the importance and reliability of this case study briefly also in the introduction, specifically lines 104-109, as follows "Although this strategic level is still missing in most of the member states [32 ], it is crucial, especially for countries located in vulnerable areas currently facing climate and pollution issues (e.g., Mediterranean region; [39]). In these countries, a wide and national perspective could help to coordinate the implementation of NBS at lower levels for reaching multiple national targets related to different environmental policies, with the final scope to improve the state of ecosystems and human health as a whole. Accordingly, we selected Italy as a case study, since it is a representative member state both for the challenges related to pollution and climate change and for the national policies in place to improve urban sustainability."

#2.2 First of all, I find that an important topic, compatible with the journal's scope, was considered.
Such studies are partially analysed in literature. It would be worth presenting the state of the art in a broader way. I suggest a more dilligent, comparative description of other scientific research from the literature (for example, it is possible to add a short state of the art comparative analysis report).

Rep: The whole study already includes 100 references, most of them used as benchmarks to delineate the current state of the art and lack in large-scale studies regarding NBS within policy discourses. We would prefer to not go further in-depth with this aspect which would be more appropriate for a review paper than for a research article like the one we are proposing.

I also recommend several corrections to improve the quality of this paper:

#2.3 - to precisely define the research scenario (it is very general); needed to clarify the scope of the study and consequently a clear, step-by-step, simple, synthetic research pattern; yes, the methodology is described, but I recommend more precision, as the reader should know how to repeat a similar analysis on this basis (please consistently correct and complete section 3);

Rep: The first paragraph of section 3 aims to present the overall methodological flow, illustrating the three main stages of the work.  We slightly modified the text to be more consistent with Figure 1 and easier introduce the readers to the understanding of the overall approach, guiding them into the following methodological steps (i.e., sections 3.1, 3.2, and 3.3). The subsequent paragraphs of the methodology now also refer to Figure 1 allowing the readers to clearly follow the main stages.

#2.4 - to improve the readability and description of figures (since they are the basis for analysis verification), supplement the history of their description, a clear and not laconic reference in the paper;

Rep: Following your suggestion, we tried to report more explicit descriptions and references to the figures in the main text in order to give them the right value and improve their readability. Example at lines 168-176; 275-277; 336-337.

#2.5 that is, supplement the discussion and summary descriptive analysis (please complete section 5).

Rep: We completed and integrated the discussion section by adding a paragraph about the work limitations and the data needed to support policymakers and planners in more effective and efficient implementation of NBS, especially on a local scale. We believe that the addition of this paragraph will help other researchers in the field replicate our methodology and identify research gaps to improve future support for multi-scale policies. You can follow our main changes on lines 371-395.

#2.6 Please remember that the formulated objectives - find a clear answer in the conclusion of the study. Is this really the way it works? Does the conclusion answer all the questions posed at the beginning of the paper (expressed in objectives and hypotheses)? Please complete it and also correct it. The conclusion needs to be supplemented (section 6). I also strongly suggest that recommendations for specific, practical, not only general (and not entirely clear) applications of this research shall be provided.

Rep: Thanks for this suggestion. In the previous version of the manuscript, the conclusions were vague and not very incisive. We slightly modified the text to report more clearly the main results obtained and explicitly link them to the objectives and hypothesis set in the introduction section. We hope that the changes made will help the reader to better understand the main outcomes and practical applications of our study. Please find the changes on lines 475-493 as follows “The environmental challenges addressed in this work can adversely affect human health and well-being, with associated mitigation costs. Accordingly, our work contains a novel framework that will help both the national government and the municipalities to identify NBS able to maximize the ES supply while addressing multiple challenges. Analogously to the already proposed “National Strategy on Urban Green-spaces” [59], this work can provide a strategic vision at the national scale but it can be consulted and adopted by all municipalities as a common roadmap, also helpful in facing the recurring problematic of planning silos. Indeed, the relevance of our framework is not just focused on the NBS application at the local scale but it shows its great impact on a wider scale (e.g., national and regional). On a national scale, the framework here proposed is reliable to i) identify the areas showing a simultaneous demand for the achievement of multiple national targets, ii) spatially orient the new investment needed to mitigate the challenges (e.g., EU Green Deal); iii) support the NBS selection that provides more co-benefits, playing a crucial role in increasing budget allocations efficiency. On a municipal scale, the NBS ranking can be used as a guideline for further specific planning and design activities based on local issues, barriers and peculiarities, while remaining consistent with national targets. ”.

#2.7 The language of this paper is relatively correct, however, some descriptions would benefit from being more concise.

Rep: thanks for your suggestions. An extensive review of the manuscript has been done to eliminate typos and inconsistencies thus hopefully enhancing its overall readability.